# Evaluation of right ventricular function during liver transplantation with transesophageal echocardiography

**Glauber Gouvêa**[1]*, **John Feiner**[2], **Sonali Joshi**[2], **Rodrigo Diaz**[3], **Jose Eduardo Ferreira Manso**[4], **Alexandra Rezende Assad**[5], **Ismar Lima Cavalcanti**[5], **Marcello Fonseca Salgado-Filho**[6], **Aline D'Avila Pereira**[7], **Nubia Verçosa**[4]

1 PostGraduate Program of Surgical Sciences, Clementino Fraga Filho University Hospital, Federal University of Rio de Janeiro, Rio de Janeiro, Rio de Janeiro, Brazil, 2 Department of Anesthesia and Perioperative Care, University of California San Francisco, San Francisco, California, United States of America, 3 Clementino Fraga Filho University Hospital, Federal University of Rio de Janeiro, Rio de Janeiro, Rio de Janeiro, Brazil, 4 Department of Surgery, PostGraduate Program of Surgical Sciences, Federal University of Rio de Janeiro, Rio de Janeiro, Rio de Janeiro, Brazil, 5 Department of General and Specialized Surgery/Anesthesiology, Fluminense Federal University, Niterói, Rio de Janeiro, Brazil, 6 PostGraduate Program in Anesthesia for High Risk Procedures, Israelita Albert Einstein Hospital, São Paulo, São Paulo, Brazil, 7 Department of Nutrition, University of Vassouras, Vassouras, Rio de Janeiro, Brazil

☯ These authors contributed equally to this work.
‡ These authors also contributed equally to this work
* glauber.gouvea@gmail.com

**Data Availability Statement:** The repository used for data sharing was Harvard Dataverse. The DOI for data assessment is https://doi.org/10.7910/DVN/G6D9G5. Only the hemodynamic data were

## Abstract

### Background

The pathophysiology of advanced liver cirrhosis may induce alterations in the circulatory system that may be challenging for the anesthesiologist to manage intraoperatively, and perioperative cardiovascular events are associated with worse outcomes in cirrhotic patients undergoing liver transplantation. It remains controversial whether right ventricular function is impaired during this procedure. Studies using transesophageal echocardiography for quantitative analysis of the right ventricle remain scarce in this setting, yielding conflicting results. The aim of this study was to perform a quantitative assessment of right ventricular function with two parameters derived from transesophageal echocardiography during liver transplantation.

### Methods

Nineteen adult patients of both genders undergoing liver transplantation were evaluated in this observational study. The exclusion criteria were age under 18 or above 65 years old, fulminant hepatic failure, hepatopulmonary syndrome, portopulmonary hypertension, cardiopulmonary disease, and contraindications to the transesophageal echocardiogram. Right ventricular function was assessed at five stages during liver transplantation: baseline, hepatectomy, anhepatic, postreperfusion, and closure by measuring tricuspid annular plane systolic excursion and right ventricular fractional area change obtained with transesophageal echocardiography.

published to keep patients' identities anonymous. Although patients' data had been collected on a consecutive basis, the hemodynamic data of each patient in the sheet were randomly scrambled to add more protection. In other words, P1 in the sheet does not necessarily mean the first patient in our cohort, and so forth.

**Funding:** The author(s) received no specific funding for this work.

**Competing interests:** The authors have declared that no competing interests exist.

## Results

Right ventricular function was found to be normal throughout the procedure. The tricuspid annular plane systolic excursion showed a trend toward a decrease in the anhepatic phase compared to baseline (2.0 ± 0.9 cm vs. 2.4 ± 0.7 cm; P = 0.24) but with full recovery after reperfusion. Right ventricular fractional area change remained nearly constant during all stages studied (minimum: 50% ± 10 at baseline and anhepatic phase; maximum: 56% ± 12 at postreperfusion; P = 0.24).

## Conclusions

Right ventricular function was preserved during liver transplantation at the time points evaluated by two quantitative parameters derived from transesophageal echocardiogram.

## Introduction

Patients with end-stage liver disease may develop perioperative cardiovascular events associated with worse outcomes [1]. This is a significant concern, particularly as the recipients' age has been increasing over the last years [2]. Thus, preoperative screening with 2D transthoracic echocardiography for all patients on the liver transplantation waiting list has been suggested by guidelines [3,4], including a dobutamine stress test in selected individuals to exclude significant myocardial ischemia. Cirrhotic cardiomyopathy is a distinct syndrome that may also be identified in preoperative screening. Typical findings include ventricular systolic or diastolic dysfunction and electrophysiological abnormalities, such as prolonged QT interval [5].

Other conditions suspected with preoperative echocardiography are hepatopulmonary syndrome and portopulmonary hypertension (PoPHTN). Regarding the latter, its pathophysiology shares some common features of primary pulmonary hypertension, including vascular remodeling [6]. It may affect 5 to 10% of patients who are candidates for a liver transplant and carries a poor prognosis if left untreated [6].

It should be noted that heart function impairment may lead to liver dysfunction, either acute or chronic. This reinforces the close physiological interplay between the heart and the liver [7]. For example, acute elevated pulmonary artery pressures may lead to right ventricular failure, which may impede liver venous outflow, leading to hepatic congestion and dysfunction. This scenario has been described after liver graft reperfusion in patients with PoPHTN undergoing orthotopic liver transplantation (OLT) [8]. In addition, major fluid shifts, bleeding, and inferior vena cava (IVC) manipulation by the surgeons often occur during the procedure, which may cause liver hypoperfusion and hemodynamic instability. These intraoperative events may overlap with the baseline hemodynamic alterations typical of cirrhotic patients, such as central hypovolemia, splanchnic hypervolemia, vasodilation, low systemic vascular resistance, and impaired vascular response to vasoconstrictors [9,10]. All of these factors are challenging for anesthesiologists to manage intraoperatively.

Although systemic hemodynamic alterations during OLT have been well reported [11], relatively few studies have focused on the right ventricular function (RVF) in this setting [12–15]. Most of these studies have used a modified pulmonary artery catheter (PAC), which measures the right ventricular ejection fraction (RVEF). More recently, the intraoperative use of transesophageal echocardiography (TEE) for cardiac function monitoring has been increasingly reported [16]. TEE has repeatedly demonstrated its usefulness during OLT, particularly in

diagnosing intracardiac thromboembolism and cardiac dysfunction [17,18]. In this regard, most clinicians use TEE for subjective assessment only, whereas quantitative analysis by echocardiography is more accurate, mainly when RVF is to be evaluated [19,20].

To date, few studies have reported using TEE to evaluate RVF quantitatively during OLT, yielding conflicting results. One [21] used 3D TEE and showed normal RVF throughout the procedure, as assessed by RVEF. Another study [22] found that right ventricular dysfunction (RVD), diagnosed either by visual estimation or using a single quantitative index, might be quite common, particularly in the anhepatic and reperfusion stages. These findings could be explained at least in part by distinct methods used by the authors to quantify the RVF.

According to current guidelines, at least two quantitative indices should be used to properly evaluate the RVF [23,24]. The tricuspid annular plane systolic excursion (TAPSE) and right ventricular fractional area change (RVFAC) are commonly used parameters. TAPSE measurement is straightforward, relatively easy to perform, has excellent intra- and interreproducibility and is an independent predictor of mortality [25,26]. RVFAC is independent of geometric assumptions and correlates well with RVEF [23,24]. TAPSE and RVFAC values less than 1.7 cm and 35%, respectively, indicate decreased RVF [24].

To the best of our knowledge, no study using TEE for RVF assessment with two quantitative parameters during OLT has been reported. Therefore, the aim of this study was to evaluate the RVF by measuring TAPSE and RVFAC obtained with TEE at five time points during OLT: baseline, hepatectomy, anhepatic, postreperfusion, and closure phases.

## Materials and methods

This study was approved by the Institutional Review Board of the Research Ethics Committee of the Clementino Fraga Filho University Hospital (HUCFF) from Faculty of Medicine of the Federal University of Rio de Janeiro (UFRJ), CAAE number 797638179.0000.5257, date of approval: 12/14/2017. The study protocol was registered at ClinicalTrials.gov as NCT03459924 (03/09/2018) and adheres to the applicable EQUATOR guidelines (STROBE checklist). Since this was a retrospective study, the Ethics Committee waived the requirement for informed consent.

We performed an observational, retrospective study and evaluated TEE data collected intraoperatively from nineteen adult patients of both genders who underwent OLT in our center. Our team has a strict protocol to save patient data and hemodynamic parameters on a database at well-defined time points throughout OLT. The principal researcher was responsible for all TEE measurements. The inclusion criteria comprised all adult patients who underwent OLT during the study period. All patients underwent a complete preoperative cardiovascular screening, including dobutamine stress echocardiography. Exclusion criteria for analysis were age under 18 or greater than 65 years old; past cardiac surgeries or cardiac disease (heart failure, arrhythmias, grade 2 or more diastolic dysfunction, left ventricular hypertrophy, significant aortic or mitral valve disease); any condition precluding the use of TEE (such as odynophagia or esophageal stricture); moderate or severe tricuspid regurgitation in the preoperative echocardiogram; fulminant liver failure; hepatopulmonary syndrome; PoPHTN, and the presence of esophageal varices with recent bleeding ($< 6$ months) or grade $> 2$.

Standard basic monitoring was initiated in the operating room with ECG leads DII and V5, noninvasive blood pressure, and a pulse oximeter (Datex AS/5 monitor, Datex GE®, Helsinki, Finland). After insertion of a peripheral intravenous line a rapid sequence induction of anesthesia was performed with fentanyl (1–3 μg kg$^{-1}$), propofol (1–2 mg kg$^{-1}$), and rocuronium (1 mg kg$^{-1}$). The endotracheal tube position was confirmed with physical examination and

capnography. A Foley catheter was inserted to monitor urine output. Neuromuscular transmission—train-of-four stimulus—was followed throughout the procedure, and the goal was to keep the train of four counts less than two. A nasopharyngeal temperature probe was carefully positioned, and air-forced warmed blankets were used to maintain normothermia (>36°C). An arterial cannula was inserted into the left radial artery for invasive blood pressure monitoring. Afterward, ultrasound-guided (M-Turbo machine–FUJIFILM SonoSite, Inc., Bothell, WA, USA) cannulation of the right internal jugular vein was performed. An 8 Fr double-lumen catheter and 8.5 Fr cordis were positioned (Arrows® International Inc., Reading, PA, USA). PAC was not used for any patient. A rapid infusion catheter (RIC 7 Fr, Arrows® International Inc., Reading, PA, USA) was inserted into the left antecubital or another large arm vein and then connected to an infusion system (Belmont®, Belmont Instrument Corporation, Boston, MA, USA). A cell saver device was used for all patients, and processed blood was returned through the rapid infusion system. Maintenance of anesthesia was performed with fentanyl (1–3 µg $kg^{-1}$ $h^{-1}$), cisatracurium (0.2–1 µg $kg^{-1}$ $min^{-1)}$ and desflurane (0.75–1.3 MAC) in an $O_2$:air mixture ($FiO_2$ = 0.5–1.0).

A 3–8 MHz TEE probe (M-Turbo machine–FUJIFILM SonoSite, Inc., Bothell, WA, USA) was positioned. After a comprehensive examination [27], the probe was secured in the mid-esophageal position to obtain the four-chamber view throughout the procedure. A dedicated anesthesia team managed the cases clinically, while the principal researcher performed all TEE examinations and recordings.

If the patient developed significant hypotension (systolic blood pressure or mean arterial pressure (MAP) less than 90 and 65 mmHg, respectively), phenylephrine 50–100 µg was administered in bolus and then a 25–100 µg $min^{-1}$ infusion. Norepinephrine 2.5–15 µg $min^{-1}$ infusion was initiated in cases of nonresponsiveness to phenylephrine. Otherwise, if hypovolemia was suspected, as assessed by hemodynamic parameters or TEE findings, fluid resuscitation was performed with albumin 5% 5 ml $kg^{-1}$ in bolus. A crystalloid infusion was kept minimal during the procedure. A fluid bolus of albumin 5% (30 ml $kg^{-1}$) was routinely given within 30 minutes before the IVC clamp unless clear signs of hypervolemia were present. Packed red blood cells were given when hemoglobin decreased below 8 g $dl^{-1}$ or if ongoing bleeding with hemodynamic instability developed. Fresh frozen plasma was administered for nonsurgical bleeding when coagulopathy was diagnosed (INR>2). Cryoprecipitate was given only in cases of severe fibrinogen deficiency (<80 mg $dl^{-1}$). Platelets were transfused in selected patients with persistent bleeding and significant thrombocytopenia (<50.000 $mm^{-3}$).

Only senior surgeons performed the surgical procedure. Most of them performed side clamping of the IVC, while some preferred a full IVC cross-clamp to make the venous anastomosis. No venovenous bypass was used in this study. If the patient developed significant hypotension after graft reperfusion (postreperfusion syndrome), epinephrine was administered (10 µg in bolus, maximum dose 50 µg), followed by vasopressin (0.2 IU in bolus) used as a rescue treatment.

RVF was assessed by two TEE-derived parameters, TAPSE and RVFAC, at five stages during the procedure: baseline (TB)—as soon as the patient developed hemodynamic stability after surgical incision; hepatectomy (TH)—approximately 10 minutes before the portal vein was clamped; anhepatic (TA)—approximately 10 minutes before the portal vein was unclamped; postreperfusion (TR) - 30 minutes after reperfusion of the graft; and closure (TC)—when abdominal wall aponeurosis closure was initiated. Other hemodynamic parameters were collected at the same stages: MAP, heart rate (HR), central venous pressure (CVP), and systolic pressure variation (SPV) (determined by using the cursor method directly on the monitor) [28].

TAPSE and RVFAC measurements were described as follows. The TEE probe was positioned in the mid-esophageal window to achieve the four-chamber view. With the right ventricle (RV) centered on the screen, the image was frozen and then captured in the end-diastole and end-systole frame using the cine loop function of the ultrasound machine. The distance (in cm) between the RV apex and the tricuspid valve annulus attached to the RV free wall was measured at end-diastole and end-systole [29]. The difference between the two values yielded the TAPSE (Fig 1). This measurement is sometimes referred to as modified TAPSE or m-TAPSE [30,31] compared to the more common method that uses the M-mode in the apical view obtained by transthoracic echocardiogram.

RVFAC was calculated by measuring the RV area with manual planimetry at end-diastole and end-systole. The following formula was then used to calculate RVFAC:

$$RVFAC = [(RVEDA - RVESA)/RVEDA]x100 \qquad (1)$$

(RVFAC: right ventricular fractional area change; RVEDA: right ventricular end-diastolic area; RVESA: right ventricular end-systolic area).

Left ventricular ejection fraction (LVEF) was calculated using the modified Simpson's biplane method [32], as TAPSE may also depend on left ventricular function [33]. The principal researcher measured all TEE parameters three times, and the average (mean value) was recorded.

## Statistical analysis

Continuous variables were assessed by Shapiro-Wilk test and summarized as mean values ± SD for normally distributed variables and median values [interquartile range; $Q_{25}$ to $Q_{75}$] for non-normally distributed variables. Frequencies and percentages were used for categorical variables. Hemodynamic and TEE data from the five stages (TB, TH, TA, TR, TC) were compared using one-way ANOVA for repeated measurements and Tukey's test was applied for post hoc analysis in case of normal distribution. Otherwise, the Friedman test was employed, and Dunns' test was applied for post hoc analysis. $P < 0.05$ was considered significant. The software used for statistical analysis was Prism 7 for Mac OS X, V7.0d (GraphPad Software, Inc., San Diego, CA, USA).

## Results

After exclusion criteria were applied, nineteen (n = 19) adult patients who underwent OLT from April 2012 to April 2013 were included for analysis. Patient characteristics and intraoperative data are shown in Table 1.

The majority of participants were male (74%). The mean age and body mass index (BMI) were 52 ± 13 years old and 27 ± 6 kg m$^{-2}$ respectively. Six patients (31%) were on low-dose beta-blocker therapy for esophageal variceal bleeding prophylaxis. The Model for End-Stage Liver Disease (MELD) score was relatively high (mean of 26), but exception points were given for six patients with hepatocellular carcinoma. Vasopressors were used in all patients at some point during the procedure. Phenylephrine infusion ranged from 25 to 80 μg min$^{-1}$ while norepinephrine ranged from 2.5 to 10 μg min$^{-1}$. Eleven (58%) patients developed postreperfusion syndrome and were successfully treated with adrenaline bolus. The median estimated blood loss was 2800 ml (IQR [1600 to 8500] ml). The total amount of fluid administered was 1894 ± 760 ml for crystalloids and 2115 ± 2062 ml for colloids (albumin 5%) (mean ± SD). The IVC was partially clamped in most patients (63%). No subjects died during the study period, and all patients were extubated in the operating room and admitted to the intensive care unit.

Hemodynamic and TEE parameters are shown in Table 2.

(A)

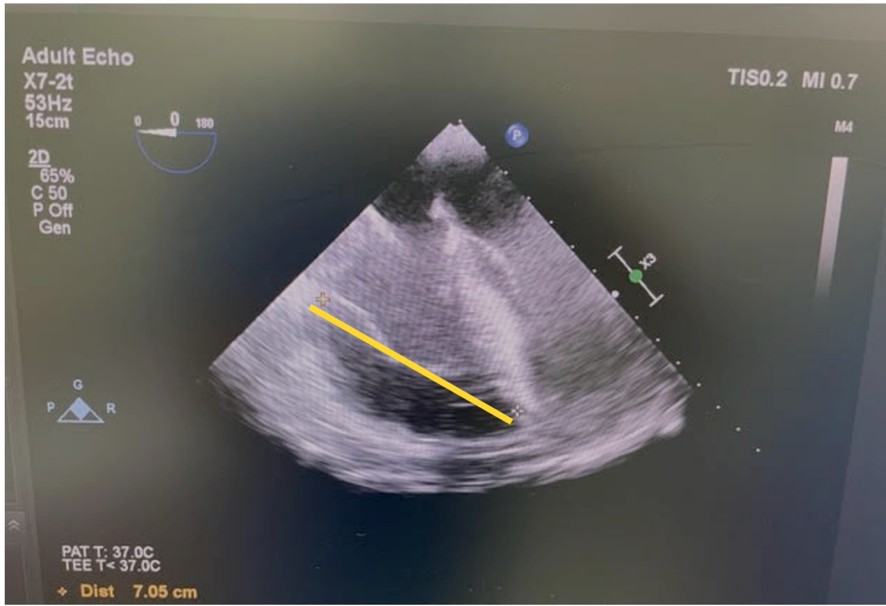

(B)

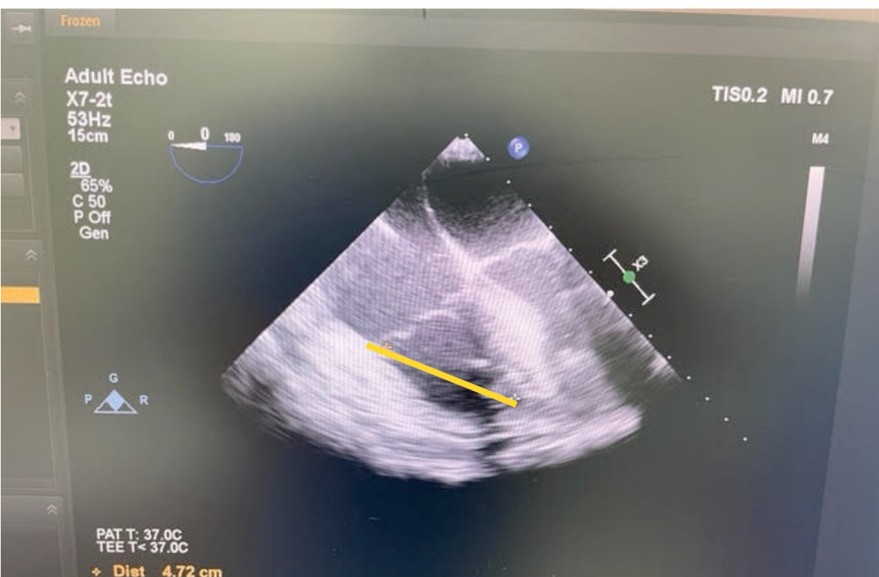

**Fig 1. TAPSE measurement.** The image of the right ventricle in the mid-esophageal four-chamber view is captured at end-diastole (A) and end-systole (B). The distance between the right ventricular apex and the lateral tricuspid annulus was measured, as shown in (A) and (B) (yellow lines: 7.05 and 4.72, respectively). The difference yields the TAPSE, in this case, 2.3 cm. Source: principal researcher's archive.

Only CVP during the anhepatic period (TA) decreased significantly from baseline (P = 0.003).

TAPSE and RVFAC were within the normal limits throughout the procedure. TAPSE showed a trend toward a decrease in the anhepatic phase (TA) when compared to baseline

**Table 1. Patient characteristics and intraoperative data.**

| | |
|---|---|
| **Gender (M/F)** | (14/5) |
| **Age (years)** | 52 ± 13 |
| **Weight (kg)** | 78 ± 14 |
| **Height (cm)** | 168 ± 9 |
| **MELD score** | 26 ± 10 |
| **Liver disease** **HCV** **Alcohol** **Biliary disease** **HBV** **Others** | 7 (37%) 3 (16%) 4 (21%) 1 (5%) 4 (21%) |
| **Preoperative drugs** **Beta-blockers** **Diuretics** | 6 (31%) 4 (21%) |
| **Length of surgery (min)** | 450 ± 112 |
| **Intraoperative fluids** **Crystalloids (ml)** **5% Albumin (ml)** | 1894 ± 760 2115 ± 2062 |
| **Diuresis (ml)** | 300 [100 to 900] |
| **Intraoperative vasopressor use** **Baseline** **Hepatectomy** **Anhepatic** **Postreperfusion** **Closure** | 9 (47%) 16 (84%) 18 (95%) 17 (89%) 14 (74%) |
| **Packed red blood cells (units)** | 4 [1.5 to 9.5] |
| **Fresh frozen plasma (units)** | 5 [0 to 11.5] |
| **Platelets (units)** | 0 [0 to 1.5] |
| **IVC clamp** **Full** **Partial (side-clamp)** | 7 (37%) 12 (63%) |
| **Postreperfusion syndrome** | 11 (58%) |
| **Estimated blood loss (ml)** | 2800 [1600 to 8500] |

Values are expressed as the absolute numbers when unspecified, means ± SD, medians [IQR 25 to 75] or n (%).
Others included nonalcoholic steatohepatitis (n = 2), cryptogenic cirrhosis (n = 1) and hemangioma (n = 1).
Abbreviations: M = male; F = female; MELD = Model for End Stage Liver Disease; HCV = Hepatitis C Virus;
HBV = Hepatitis B Virus; IVC = Inferior vena cava.

(TB) (mean ± SD: 2.0 ± 0.9 cm vs. 2.4 ± 0.7 cm; P = 0.24) but fully recovered after reperfusion. RVFAC remained nearly constant during all phases studied (minimum: 50% ± 10 at TB and TA; maximum: 56% ± 12 at TR; P = 0.24).

## Discussion

In this study, RVF was preserved during OLT, as revealed by TAPSE and RVFAC measurements obtained by TEE. Only TAPSE showed a mild trend toward a decrease in the anhepatic phase, with full recovery after reperfusion. This biphasic pattern of RVF has also been shown in previous studies using a modified PAC, which measures RVEF [12–15]. Indeed, TAPSE and RVEF correlate well when measured by cardiac magnetic resonance imaging [34], and our results reinforce this relationship.

The mean TAPSE and RVFAC found in our study at baseline were 2.4 cm and 50%, respectively, which should be considered normal. However, the normal range in this patient

**Table 2. Intraoperative hemodynamic and TEE data.**

| Variable | TB | TH | TA | TR | TC | P—value |
|---|---|---|---|---|---|---|
| **MAP (mmHg)** | 76 [62 to 84] | 85 [70 to 95] | 73 [69 to 80] | 70 [63 to 73] | 70 [67 to 78] | 0.15 |
| **HR (bpm)** | 73 ± 11 | 79 ± 15 | 81 ± 19 | 77 ± 16 | 78 ± 13 | 0.08 |
| **CVP (mmHg)** | 12 ± 7 | 8 ± 5 | 5 ± 6* | 9 ± 4 | 9 ± 5 | 0.003 |
| **SPV (mmHg)** | 4 [3 to 6] | 4 [3 to 8] | 7 [4 to 15] | 4 [3 to 7] | 4 [2 to 10] | 0.10 |
| **TAPSE (cm)** | 2.4 ± 0.7 | 2.4 ± 0.9 | 2.0 ± 0.9 | 2.5 ± 0.5 | 2.7 ± 0.9 | 0.24 |
| **RVFAC (%)** | 50 ± 10 | 55 ± 13 | 50 ± 11 | 56 ± 12 | 52 ± 12 | 0.24 |
| **LVEF (%)** | 66 ± 12 | 70 ± 14 | 68 ± 12 | 71 ± 12 | 70 ± 12 | 0.82 |

Data are expressed as the means ± SD or medians [IQR 25–75%] when appropriate. Abbreviations: TB: Baseline; TH: Hepatectomy stage; TA: Anhepatic stage; TR: Postreperfusion stage; TC: Closure stage; MAP: Mean systemic arterial pressure; HR: Heart rate; CVP: Central venous pressure; SPV: Systolic pressure variation; TAPSE: Tricuspid annular plane systolic excursion; RVFAC: Right ventricular fractional area change; LVEF: Left ventricular ejection fraction. Statistical comparisons were by repeated-measures ANOVA or Friedman's test

*$P < 0.05$ from baseline.

population is still unclear. One study reported similar values between cirrhotic and healthy individuals [35]. In contrast, two other studies [36,37] found that TAPSE was significantly higher among cirrhotic patients. If the latter data were considered, our patients' TAPSE and RVFAC baseline values would be interpreted as decreased. However, the reproducibility of these findings remains to be determined. It is possible that in some patients with liver failure, the hyperdynamic circulation may help preserve TAPSE values in the normal range. Thus, more studies should be performed to identify a different baseline for hyperdynamic cirrhotic patients.

TAPSE is influenced by the loading conditions of the RV, particularly preload [38]. Of note, the mild alterations in CVP and SPV in the anhepatic phase found in our study suggest that the RV preload was not markedly reduced. This was likely due to the partial IVC clamp performed more frequently and the large fluid bolus of albumin routinely given before clamping. Furthermore, 95% of the patients were under vasopressors during this phase. All those factors might have contributed to the transient and slight TAPSE reduction found in this phase. This contrasts with the study of Shillcutt et al. [22], where a full IVC cross-clamp was routinely used for all patients. The incidence of RVD was 15% in the anhepatic phase, as assessed by TAPSE or visual estimation. However, the visual assessment of RVF is known to be inaccurate [23,24]. Therefore, RVD could have been overestimated. Indeed, those authors acknowledged the unusually high incidence of RVD compared to other studies [12–15,21]. Furthermore, fluid management data are lacking, and the authors did not clearly define the exact time point where measurements had been taken. Conversely, similar to our findings, Rosendal et al. [21] reported that RVEF was nearly unchanged in the anhepatic phase, as assessed by 3D-TEE. In their study, all patients received a partial IVC clamp, suggesting that RV preload may significantly influence RVF.

A decreased TAPSE may be secondary to left ventricular dysfunction or tachycardia [33], but neither LVEF nor HR showed significant alterations in our patients. In contrast to the normal LVEF values found in our study, cirrhotic patients may have a lower LVEF [5]. In our center, however, most of our patients had relatively stable liver disease, and preoperative left ventricular dysfunction was unusual. In addition, norepinephrine was frequently used intraoperatively, and its inotropic effects could have contributed. Changes in RV afterload were also unlikely to result in a decreased TAPSE, as previous studies have shown that pulmonary vascular resistance usually remains unchanged or slightly decreases in the anhepatic phase [14,15].

We do not know how the pulmonary pressure changed as we did not use PAC. The degree of tricuspid insufficiency may worsen in the face of an increase in pulmonary pressure or during intraoperative maneuvers, with consequent maintenance of TAPSE [39]. Finally, decreased RV contractility leading to a reduced TAPSE cannot be excluded, as it has been shown that myocardial depressant factors accumulate during OLT [40].

The RVFAC did not follow TAPSE trends in the anhepatic phase. Two possible reasons could explain this: first, RVFAC measurement is less reproducible than TAPSE, showing more significant variability among intra- or interobservers [29,41]. Second, preload changes may have a more considerable influence on TAPSE readings when compared to RVFAC [23]. The normal RVEF found in Rosendal et al. [21]'s study agrees with our results, as RVFAC may be considered a two-dimensional surrogate for RVEF [23,24].

Ellis et al. [42] used TEE during OLT and reported some cases of significant RVD within five minutes after reperfusion. Their results should be interpreted with caution, as a quantitative analysis was not performed. Moreover, surgical and anesthetic management for liver transplantation has dramatically evolved since their study. Shillcutt et al. [22] found a 22% incidence of RVD after reperfusion. However, the measurement timing was not standardized, and the authors neither quantified the degree of RV impairment nor reported the TAPSE values. Rosendal et al. [21] found a normal RVEF measured by 3D-TEE five minutes after reperfusion, suggesting that RVF was not impaired at this time. This agrees with the present study, as RVF assessment during this phase also showed normal values.

The timing of RVF assessment after reperfusion is of great relevance and influences the results. Unlike the authors above [21,22,42], who evaluated the RVF within five minutes, we assessed RVF thirty minutes postreperfusion. This time point was chosen for two reasons: first, the five minutes after reperfusion is a stressful and demanding time for the anesthesiologist. Even for experienced clinicians, performing quantitative and precise TEE measurements may be challenging during this time. Second, any RVD eventually found at this time point would be unrelated to the acute hemodynamic and metabolic events following five minutes of reperfusion. Accordingly, 58% of our patients received adrenaline bolus to overcome the postreperfusion syndrome, and this drug would influence RVF acutely.

Overall, our results are similar to those of Rosendal et al. [21]. However, in their study, only one parameter (RVEF) was measured as an index for RVF. At the same time, it has been recommended that at least two parameters be used to properly quantify the RVF [23,24]. Accordingly, both TAPSE and RVFAC were measured in our study, suggesting that RVF is well preserved throughout OLT when measured at specific time points during the procedure.

Further studies using more accurate indices of RVF, such as tissue Doppler and global strain of the RV, may be warranted to define whether clinically significant RVD occurs intraoperatively and whether it influences patient outcomes.

## Study limitations

This was a retrospective study with a small number of patients enrolled. Thus, a type 2 statistical error cannot be excluded. The accuracy of echocardiographic measurements could have been improved if two or three physicians (blinded to each other) performed the exam. However, to accomplish this, those measurements should ideally have been done offline, but unfortunately, we did not have any specific software available to do so. Although we did not use PAC routinely, it could likely add more information about RVF, such as the pulmonary artery pulsatility index (pulmonary artery pulse pressure divided by the CVP) [43]. More accurate and less preload-dependent indices of RVF could have been measured, such as the global longitudinal strain and tissue Doppler [44]. Fluid therapy may influence RVF, and although we

have reported the total amount of crystalloids and colloids during surgery, we lacked data on total fluid balance. Finally, we collected data from patients who underwent relatively uncomplicated liver transplants. Our results cannot be extrapolated to specific conditions, such as massive transfusion or profound vasoplegia, and patients with PoPHTN or hepatopulmonary syndrome who might behave differently.

## Conclusions

In our study, RVF was preserved during OLT when evaluated by quantitative analysis with TAPSE and RVFAC derived from TEE, as assessed at five stages: baseline, hepatectomy, anhepatic, postreperfusion, and closure.

## Acknowledgments

We are grateful for the enormous support and mentorship of Claus U Niemann, MD, PhD, Professor of Surgery, member of the Liver Transplant Team of the Moffit Long Hospital, University of California San Francisco (UCSF), USA.

## Author Contributions

**Conceptualization:** Glauber Gouvêa, John Feiner, Sonali Joshi, Rodrigo Diaz.

**Data curation:** Glauber Gouvêa, John Feiner, Sonali Joshi, Rodrigo Diaz.

**Formal analysis:** Glauber Gouvêa, John Feiner, Sonali Joshi, Rodrigo Diaz.

**Investigation:** Glauber Gouvêa, John Feiner, Sonali Joshi, Rodrigo Diaz.

**Methodology:** Glauber Gouvêa, John Feiner, Sonali Joshi, Rodrigo Diaz.

**Supervision:** Jose Eduardo Ferreira Manso, Alexandra Rezende Assad, Ismar Lima Cavalcanti, Marcello Fonseca Salgado-Filho, Aline D'Avila Pereira, Nubia Verçosa.

**Validation:** Jose Eduardo Ferreira Manso, Alexandra Rezende Assad, Ismar Lima Cavalcanti, Marcello Fonseca Salgado-Filho, Aline D'Avila Pereira, Nubia Verçosa.

**Visualization:** Jose Eduardo Ferreira Manso, Alexandra Rezende Assad, Ismar Lima Cavalcanti, Marcello Fonseca Salgado-Filho, Aline D'Avila Pereira, Nubia Verçosa.

**Writing – original draft:** Glauber Gouvêa, John Feiner, Sonali Joshi, Rodrigo Diaz, Jose Eduardo Ferreira Manso, Alexandra Rezende Assad, Ismar Lima Cavalcanti, Marcello Fonseca Salgado-Filho, Aline D'Avila Pereira, Nubia Verçosa.

**Writing – review & editing:** Glauber Gouvêa, John Feiner, Sonali Joshi, Rodrigo Diaz, Jose Eduardo Ferreira Manso, Alexandra Rezende Assad, Ismar Lima Cavalcanti, Marcello Fonseca Salgado-Filho, Aline D'Avila Pereira, Nubia Verçosa.

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
