## [Decision Letter · Decision Letter 0]

18 Jul 2022

PONE-D-22-12798EVALUATION OF RIGHT VENTRICULAR FUNCTION DURING LIVER

TRANSPLANTATION WITH TRANSESOPHAGEAL ECHOCARDIOGRAPHYPLOS ONE

Dear Dr. Gouvêa,

Thank you for submitting your manuscript to PLOS ONE. After careful consideration, we feel that it has merit but does not fully meet PLOS ONE’s publication criteria as it currently stands. Therefore, we invite you to submit a revised version of the manuscript that addresses the points raised during the review process. The authors should improve the quality of their manuscript by addressing all issues raised by expert reviewers and by editing the english and grammar of the text.

We look forward to receiving your revised manuscript.

Kind regards,

Vincenzo Lionetti, M.D., PhD

Academic Editor

PLOS ONE

Journal Requirements:

Reviewers' comments:

Reviewer's Responses to Questions

**Comments to the Author**

1. Is the manuscript technically sound, and do the data support the conclusions?

Reviewer #1: Partly

Reviewer #2: Yes

2. Has the statistical analysis been performed appropriately and rigorously? 

Reviewer #1: No

Reviewer #2: Yes

3. Have the authors made all data underlying the findings in their manuscript fully available?

Reviewer #1: No

Reviewer #2: Yes

4. Is the manuscript presented in an intelligible fashion and written in standard English?

Reviewer #1: Yes

Reviewer #2: Yes

5. Review Comments to the Author

Reviewer #1: Cardiac complications in liver disease, especially in cirrhosis, are associated with increased mortality following liver transplant. Pre-transplant guidelines recommend risk stratification with 2D and dobutamine stress echocardiography, which may lead to coronary angiography or right heart catheterization. There is a paucity of data on the utility of RV hemodynamics during the perioperative period.

Cardiovascular events have a major impact on the outcomes of liver transplantation, especially since contemporary liver transplant patients are older than their predecessors and more likely to have cardiac comorbidities. Additionally, the pathophysiologic effects of advanced liver disease on the circulatory system pose challenges to perioperative management in liver transplant.

2012 American College of Cardiology and American Heart Association (ACC/AHA) guidelines for evaluation of cardiac disease in kidney and liver transplant patients note that it is reasonable for patients to undergo echocardiography to assess for pulmonary hypertension and intrapulmonary arteriovenous shunting, while 2014 guidelines from the American Association for the Study of Liver Diseases (AASLD) and American Society of Transplantation (AST) note that echo for this purpose should be done routinely. Portopulmonary hypertension is found in 5% to 10% of patients with chronic liver disease. Unless patients undergo liver transplant or start appropriate medical therapy, portopulmonary hypertension carries a very poor prognosis.

The pathophysiologic mechanisms specific to PoPHTN have been compared with other known forms of pulmonary hypertension, including primary pulmonary hypertension, and has been found to fall within a spectrum of disorders related to factors both due to intrinsic liver failure [with resultant portal hypertension and hepatopulmonary syndrome (HPS)] as well as pulmonary vascular remodeling.

The heart and the liver are in close relation to each other. Impairment of cardiac function may lead to hepatic dysfunction and vice versa. Liver hypoperfusion and hepatic congestion are the 2 central pathophysiological mechanisms, both in acute cardiogenic liver injury and hepatic congestion. Cirrhotic cardiomyopathy is a syndrome that includes systolic, diastolic, and electrophysiological abnormalities that develop in the setting of liver cirrhosis.

A significant number of patients presenting for liver transplant carry hemodynamic sequelae of end-stage liver disease, including generalized vasodilation, low systemic vascular resistance and an impaired vasoconstrictive response to endogenous and exogenous vasoconstrictors. These patients also have simultaneous central hypovolemia with splanchnic hypervolemia. The combination of acute blood loss, large fluid shifts and manipulation of the inferior vena cava during surgery can put significant stress on the cardiovascular system. Because of these factors, intraoperative hemodynamic instability is common during the dissection phase of liver transplant (due to blood loss) and the hepatic phase (due to obstruction of the inferior vena cava).

Mean normal TAPSE and PAPI in the literature are 2.0cm and 2.75, respectively, but in many studies mean TAPSE was 2.52cm and PAPI was 3.54. This is likely explained by the high cardiac output state in cirrhosis. In the RV failure group, TAPSE and PAPI were within the normal range suggesting the need to identify a different baseline for patients with a hyperdynamic state due to cirrhosis.

RV function was impaired in patients with cirrhosis and more commonly, in patients with ascites. However, values of RV-GLS did not distinguish the degree of severity of liver disease. In addition, the LVEF was low and LV-GLS was normal in patients with cirrhosis.

The manuscript contains many critical points. As stated by the authors, the number of enrolled patients is low and the "normal" haemodynamic conditions of the patients place this group as not representative of the average liver transplant population.

Other significant data is linked to the hyper dynamism of patients with hepatic insufficiency which can lead to a maintenance of TAPSE values in the normal range.

It would be more appropriate to study a GL strain of the RV.

The manuscript lacks data on fluid therapy (quantity and water balance) and pharmacological treatment (quantity of vasoconstrictor, if repeated boluses or continuous infusion).

We do not know if it was present and how the pulmonary pressure changed; the degree of tricuspid insufficiency present was mild and this situation may worsen during intraoperative maneuvers with consequent maintenance of TAPSE, in the face of an increase in pulmonary pressure.

The manuscript is interesting and I agree with extending intraoperative echocardiographic evaluation to most liver transplant candidates. I would also suggest to evaluate, in addition to the parameters reported in the study, also the tissue doppler and the global strain of the right ventricle.

Reviewer #2: The authors are to be commended for this study of RV function during liver transplantation. The study is well done, acknowledges prior studies on this subject appropriately, and adds to the body of evidence concerning monitoring of RV function throughout the operative period of liver transplantation.

The manuscript flows well, no significant grammatical errors (although authors should read the final version closely to eliminate any grammar problems), and is scientifically rigorous.

I wanted to state the method used for TAPSE is the m-TAPSE but this was noted in the methods section. One consideration is to explain this earlier (such as the introduction) but I don't think it's absolutely necessary.

The common surgical method for liver transplant anastomosis - partial caval clamp - was performed in the study which makes the findings of RV assessment that much more relevant to current practice of intraoperative monitoring.

6. PLOS authors have the option to publish the peer review history of their article (what does this mean?). If published, this will include your full peer review and any attached files.

Reviewer #1: No

Reviewer #2: **Yes: **Antolin S. Flores MD

---

## [Author Response · Author response to Decision Letter 0]

29 Aug 2022

Reviewer #1

Cardiac complications in liver disease, especially in cirrhosis, are associated with increased mortality following liver transplant. Pre-transplant guidelines recommend risk stratification with 2D and dobutamine stress echocardiography, which may lead to coronary angiography or right heart catheterization.

We agree with your comments. All patients in our Liver Center undergo a complete cardiovascular screening, including dobutamine stress echocardiography (DSE) if indicated. In our cohort, all of them had a normal DSE. We included this information in the Methods. We also added this information in the first paragraph of the Introduction. References number 1 to 4 were added accordingly. 

There is a paucity of data on the utility of RV hemodynamics during the perioperative period.

Indeed, this is one reason we have been focusing on RV monitoring in this setting. This has been written in the Introduction (4th paragraph).

Cardiovascular events have a major impact on the outcomes of liver transplantation, especially since contemporary liver transplant patients are older than their predecessors and more likely to have cardiac comorbidities. Additionally, the pathophysiologic effects of advanced liver disease on the circulatory system pose challenges to perioperative management in liver transplant.

2012 American College of Cardiology and American Heart Association (ACC/AHA) guidelines for evaluation of cardiac disease in kidney and liver transplant patients note that it is reasonable for patients to undergo echocardiography to assess for pulmonary hypertension and intrapulmonary arteriovenous shunting, while 2014 guidelines from the American Association for the Study of Liver Diseases (AASLD) and American Society of Transplantation (AST) note that echo for this purpose should be done routinely. 

Our Liver Center also follows these guidelines. All patients on the waiting list undergo screening with echocardiography for portopulmonary hypertension (PoPHTN) and intrapulmonary shunting (particularly in patients with a history of hypoxemia). We added this information to the first and second paragraphs of the Introduction and referenced it accordingly (references number 3 and 4).

Portopulmonary hypertension is found in 5% to 10% of patients with chronic liver disease. Unless patients undergo liver transplant or start appropriate medical therapy, portopulmonary hypertension carries a very poor prognosis.

The pathophysiologic mechanisms specific to PoPHTN have been compared with other known forms of pulmonary hypertension, including primary pulmonary hypertension, and has been found to fall within a spectrum of disorders related to factors both due to intrinsic liver failure [with resultant portal hypertension and hepatopulmonary syndrome (HPS)] as well as pulmonary vascular remodeling.

We have excluded patients with PoPHTN in our study but fully agree with your comments. We believe these patients should be studied more precisely in multicenter research, as PoPHTN occurs less commonly. We highlighted information about patients with PoPHTN in the Introduction (second paragraph) and added reference number 6. 

The heart and the liver are in close relation to each other. Impairment of cardiac function may lead to hepatic dysfunction and vice versa. Liver hypoperfusion and hepatic congestion are the 2 central pathophysiological mechanisms, both in acute cardiogenic liver injury and hepatic congestion. 

We added this information in the third paragraph of the Introduction. References 7 and 8 were included. 

Cirrhotic cardiomyopathy is a syndrome that includes systolic, diastolic, and electrophysiological abnormalities that develop in the setting of liver cirrhosis.

We agree that the role of cirrhotic cardiomyopathy cannot be overemphasized. We added this information in the Introduction (first paragraph) and reference number 5. 

A significant number of patients presenting for liver transplant carry hemodynamic sequelae of end-stage liver disease, including generalized vasodilation, low systemic vascular resistance and an impaired vasoconstrictive response to endogenous and exogenous vasoconstrictors. These patients also have simultaneous central hypovolemia with splanchnic hypervolemia. The combination of acute blood loss, large fluid shifts and manipulation of the inferior vena cava during surgery can put significant stress on the cardiovascular system. Because of these factors, intraoperative hemodynamic instability is common during the dissection phase of liver transplant (due to blood loss) and the hepatic phase (due to obstruction of the inferior vena cava).

Hemodynamic data were collected during specific time points, such as hepatectomy (dissection phase), anhepatic, and post-reperfusion phases. Cardiovascular function (including RVF) could likely be compromised at these phases. The type of inferior vena cava clamp (partial versus full) was also described in our paper. All this critical information has been added in the Introduction (third paragraph) and referenced (numbers 9 and 10). 

Mean normal TAPSE and PAPI in the literature are 2.0cm and 2.75, respectively, but in many studies mean TAPSE was 2.52cm and PAPI was 3.54. This is likely explained by the high cardiac output state in cirrhosis. In the RV failure group, TAPSE and PAPI were within the normal range suggesting the need to identify a different baseline for patients with a hyperdynamic state due to cirrhosis.

This information has been reinforced in the second paragraph of the Discussion section. 

We highlighted that more studies should be done to define the normal range for cirrhotic patients. 

Unfortunately, we did not measure the PAPI, calculated by pulmonary artery pulse pressure divided by the central venous pressure. This is a valuable RV function index that has been used in some recent studies. We added this in the Discussion (study limitations).

RV function was impaired in patients with cirrhosis and more commonly, in patients with ascites. However, values of RV-GLS did not distinguish the degree of severity of liver disease. In addition, the LVEF was low and LV-GLS was normal in patients with cirrhosis.

Unfortunately, we did not measure the degree of ascites in our patients. However, none had severe ascites as observed clinically or during surgery. This may be explained by our selective cohort of relatively stable patients. 

The RV-GLS is a more accurate parameter to evaluate the right ventricular function, and less preload-dependent. We also included this in the study limitations. We look forward to measuring it in future studies. 

In our results, LVEF was normal throughout the procedure. Most of our patients were under norepinephrine intraoperatively, and its inotropic effects could possibly explain a higher than expected LVEF. We added this comment in the Discussion section (4th paragraph). 

The manuscript contains many critical points. As stated by the authors, the number of enrolled patients is low, and the "normal" haemodynamic conditions of the patients place this group as not representative of the average liver transplant population.

The authors agree and have pointed out these issues in the Discussion section (study limitations). Further studies with more severely affected patients should be done.

Other significant data is linked to the hyper dynamism of patients with hepatic insufficiency, which can lead to a maintenance of TAPSE values in the normal range.

This observation was inserted into the Discussion, second paragraph.

It would be more appropriate to study a GL strain of the RV.

We agree that more accurate parameters should be used to evaluate the RV. The GL strain of the RV is certainly one, and this issue has been added into the Discussion (last paragraph).

The manuscript lacks data on fluid therapy (quantity and water balance) and pharmacological treatment (quantity of vasoconstrictor, if repeated boluses or continuous infusion).

In our study, only the total amount of crystalloids (mean of 1894 ml) and albumin (mean of 2115 ml) throughout the procedure were reported (Table 1). We presented only generalized data about the use or not of vasopressors in different phases of the transplant (Table 1), highlighted as “Intraoperative vasopressor use .” The role of vasopressors influencing RVF is described in the Discussion. 

We do not know if it was present and how the pulmonary pressure changed; the degree of tricuspid insufficiency was mild. This situation may worsen during intraoperative maneuvers with consequent maintenance of TAPSE in the face of an increase in pulmonary pressure.

Pulmonary artery catheter data would undoubtedly add more information to our paper, but unfortunately, we do not insert them on a routine basis anymore. We added this thoughtful observation in the Discussion section. Reference number 38 was added.

The manuscript is interesting and I agree with extending intraoperative echocardiographic evaluation to most liver transplant candidates. I would also suggest to evaluate, in addition to the parameters reported in the study, also the tissue doppler and the global strain of the right ventricle.

Thank you. Your suggestion is actually in our plans. We are willing to purchase an advanced echocardiogram 

software package to extract recorded images during the procedure. This will allow us to measure various parameters to assess the RVF more accurately. The global strain and tissue Doppler will undoubtedly be part of it. We added this in the discussion (conclusions).

We included a new coauthor who revised the statistical analysis.

We thanked you so much for your thoughtful comments and added them accordingly to the paper. 

Reviewer #2

The authors are to be commended for this study of RV function during liver transplantation. The study is well done, acknowledges prior studies on this subject appropriately, and adds to the body of evidence concerning monitoring of RV function throughout the operative period of liver transplantation.

We appreciate your comments; thank you.

The manuscript flows well, no significant grammatical errors (although authors should read the final version closely to eliminate any grammar problems), and is scientifically rigorous.

We are committed to reading the final version carefully to avoid any grammar issues. 

I wanted to state the method used for TAPSE is the m-TAPSE but this was noted in the methods section. One consideration is to explain this earlier (such as the Introduction) but I don't think it's absolutely necessary.

We also thought to include m-TAPSE earlier in the text, but in the end, we decided to put it in the Methods, where a more detailed description of the echocardiogram measurements is given. We thought it could be a bit confusing to the reader if it had been written at the beginning of the text. Thus, we will keep the m-TAPSE description in the Methods when sending the revision. However, should the reviewer change his mind and think it would be better in the Introduction, we will change it accordingly. 

The common surgical method for liver transplant anastomosis - partial caval clamp - was performed in the study which makes the findings of RV assessment that much more relevant to the current practice of intraoperative monitoring.

We agree with this point. Partial clamping of the inferior venous cava is performed much more commonly nowadays.

Thank you so much for your thoughtful comments, and we will keep following your suggestions in future revisions if you perform it again.

---

## [Decision Letter · Decision Letter 1]

13 Sep 2022

Evaluation of right ventricular function during liver transplantation with transesophageal echocardiography

PONE-D-22-12798R1

Dear Dr. Gouvêa,

We’re pleased to inform you that your manuscript has been judged scientifically suitable for publication and will be formally accepted for publication once it meets all outstanding technical requirements.

Kind regards,

Vincenzo Lionetti, M.D., PhD

Academic Editor

PLOS ONE

Additional Editor Comments (optional):

Reviewers' comments:

Reviewer's Responses to Questions

**Comments to the Author**

1. If the authors have adequately addressed your comments raised in a previous round of review and you feel that this manuscript is now acceptable for publication, you may indicate that here to bypass the “Comments to the Author” section, enter your conflict of interest statement in the “Confidential to Editor” section, and submit your "Accept" recommendation.

Reviewer #1: All comments have been addressed

Reviewer #2: All comments have been addressed

2. Is the manuscript technically sound, and do the data support the conclusions?

Reviewer #1: Yes

Reviewer #2: Yes

3. Has the statistical analysis been performed appropriately and rigorously? 

Reviewer #1: Yes

Reviewer #2: Yes

4. Have the authors made all data underlying the findings in their manuscript fully available?

Reviewer #1: Yes

Reviewer #2: Yes

5. Is the manuscript presented in an intelligible fashion and written in standard English?

Reviewer #1: Yes

Reviewer #2: Yes

6. Review Comments to the Author

Reviewer #1: Thank you very much for addressing every questions. In this revised version, the manuscript is clearer in understanding the cardiovascular issues in liver transplant patients. Some points deserve more elucidation, but in consideration of the understanding of the text, they would not produce substantial improvements.

Reviewer #2: Thank you for the revision of your manuscript. The discussion section is greatly improved and this study adds to the body of evidence regarding RV function and monitoring of it during liver transplantation. I didn't find any large grammar errors and the conclusion discusses all of the results appropriately. It flows well and doesn't make assumptions that aren't supported by data - it is scientifically rigorous.

I hope that the authors consider a larger study in the future or a multicenter study.

7. PLOS authors have the option to publish the peer review history of their article (what does this mean?). If published, this will include your full peer review and any attached files.

Reviewer #1: **Yes: **luigi tritapepe

Reviewer #2: No

---

## [Editor Report · Acceptance letter]

26 Sep 2022

PONE-D-22-12798R1 

Evaluation of right ventricular function during liver transplantation with transesophageal echocardiography 

Dear Dr. Gouvêa:

I'm pleased to inform you that your manuscript has been deemed suitable for publication in PLOS ONE. Congratulations! Your manuscript is now with our production department. 

Kind regards, 

on behalf of

Prof. Vincenzo Lionetti 

Academic Editor

PLOS ONE